# Sensor Positioning Influences the Accuracy of Knee Rom Data of an E-Rehabilitation System: A Preliminary Study with Healthy Subjects

**DOI:** 10.3390/s20082237

**Published:** 2020-04-15

**Authors:** Carlos J. Marques, Christian Bauer, Dafne Grimaldo, Steffen Tabeling, Timo Weber, Alexander Ehlert, Alexandre H. Mendes, Juergen Lorenz, Frank Lampe

**Affiliations:** 1Science Office of the Orthopedic and Joint Replacement Department, Schoen Clinic Hamburg Eilbek, Dehnhaide 120, D-22081 Hamburg, Germany; 2Faculty of Life Sciences at the Hamburg University of Applied Sciences, Ulmenliet 19, D-21033 Hamburg, Germany

**Keywords:** telerehabilitation, e-Health, physiotherapy, range of movement, knee, validity, total knee arthroplasty

## Abstract

E-rehabilitation is the term used to define medical rehabilitation programs that are implemented at home with the use of information and communication technologies. The aim was to test whether sensor position and the sitting position of the patient influence the accuracy of knee range of movement (ROM) data displayed by the BPMpathway e-rehabilitation system. A preliminary study was conducted in a laboratory setting with healthy adults. Knee ROM data was measured with the BPMpathway e-rehabilitation system and simultaneously with a BIOPAC twin-axis digital goniometer. The main outcome was the root mean squared error (RMSE). A 20% increase or reduction in sitting height led to a RMSE increase. A ventral shift of the BPMpathway sensor by 45° and 90° caused significant measurement errors. A vertical shift was associated with a diminution of the measurement errors. The lowest RMSE (2.4°) was achieved when the sensor was placed below the knee. The knee ROM data measured by the BPMpathway system is comparable to the data of the concurrent system, provided the instructions of the manufacturer are respected concerning the sitting position of the subject for knee exercises, and disregarding the same instructions for sensor positioning, by placing the sensor directly below the knee.

## 1. Introduction

Total knee (TKA) and hip arthroplasty (THA) are considered the treatments of choice for osteoarthritis patients with pain and substantial functional impairments who have not achieved acceptable pain relief and functional improvement after conservative treatment modalities [1,2]. As the population in different countries ages, there is an expected increase in the demand for THA and TKA procedures in the coming decades [3,4]. This increasing demand will present the healthcare systems and health care providers with new challenges.

After TKA and THA, physiotherapy and exercise lead to improvements in physical function [5,6,7,8]. Physiotherapy exercise provided at home is an appealing approach with the possibility of wider acceptability and uptake in the near future [9]. Studies have shown that standardized home programs were not inferior to inpatient or outpatient physiotherapy in regard to their effects on functional status, health related quality of life (HRQOL) and range of movement (ROM) of the operated joint [10,11,12,13].

E-rehabilitation or telerehabilitation are the terms used to define medical rehabilitation programs that are implemented at home with the use of information and communication technologies (ICT) [14]. Home exercise programs with the use of telerehabilitation before and after TKA and THA seem not to be inferior to standard physiotherapy in terms of improvements in range of movement and muscle strength, decrease in pain, patient satisfaction, and patient reported outcome measures (PROMs) [15,16,17,18,19,20].

The BPMpathway (270 Vision Ltd., Andover, UK) is an online, home-based biofeedback and remote patient monitoring system. The system comprises a single high-performance wearable wireless sensor, a patient application that can be downloaded for free, and a software application for the clinic. The sensor streams data to the patient application, displaying the real-time results on an animated avatar enabling the patient to visualize his/her ROM during the exercise (biofeedback) and to compare the achievements between the training sessions. The system can be used for knee, hip, and shoulder exercises.

The ROM data displayed by the BPMpathway system is calculated based on the signals of a single sensor that, according to the instructions of the manufacturer, should be placed above the malleolus lateralis, when the patient intends to perform knee exercises.

The aim of the present preliminary study was to test whether patients sitting position and the position of the sensor on the leg influence the accuracy of ROM data displayed by the BPMpathway system for knee exercises.

## 2. Materials and Methods

### 2.1. Study Design and Sample

This is a preliminary study carried out in a laboratory setting. The study was planned and developed in accordance with the “Guidelines for Reporting Reliability and Agreement Studies” (GARRAS) [21]. The reporting follows the STROBE Statement guidelines for reporting observational studies [22].

The ROM data obtained with the BPMpathway system were compared with the ROM data acquired simultaneously with the use of a twin-axis digital goniometer (BIOPAC Systems, Inc., California, USA).

The effects of sensor-related (position of the sensor on the limb) and ergonomic factors (chair height) on ROM data obtained with the BPMpathway were investigated.

The sample consisted of three healthy male subjects. The subjects were informed about the study and asked to participate. Before participating, the subjects were required to read and sign an informed consent form. The ethics commission of the Health Competence Center at the Hamburg University of Applied Sciences, Hamburg, Germany, approved the research project.

### 2.2. The BPMpathway System

The BPMpathway system comprises a single wearable wireless sensor (Figure 1A), a software application for the patients (Figure 1C) and a software program for the clinic. During the hospital stay the physiotherapist selects specific exercises/tests for an individual patient and teaches the patient how to use the system. During the test at home, the sensor streams data to the software application displaying live results on an animated avatar (biofeedback). The data is also recorded for immediate graphical comparison to previous sessions. Once connected with the internet, the data are transmitted through a cloud-based service to the physiotherapist or clinician. This ensures the progress and recovery of the patient is monitored. The patient and physiotherapist can communicate through the system and the rehabilitation program can be modified or complemented at any time by the physiotherapist at the clinic with no need of a patient visit.

### 2.3. The BIOPAC Goniometer

A twin-axis goniometer (Twin-axis Goniometer 150, TSD130B, BIOPAC Systems, Inc., California, CA, USA) was connected wirelessly to a BIOPAC MP160 unit (BIOPAC Systems, Inc., California, CA, USA) and used to measure ROM simultaneously during the tests.

### 2.4. Procedures

The goniometer was calibrated according to the instructions of the manufacturer and attached to the lateral side of the knee. To place the digital goniometer, bony landmarks were used as described in the literature [23]. To avoid dislocation of the goniometer during testing, it was attached to the leg with tape and Velcro stripes. The BPMpathway sensor was attached to the leg with a single Velcro strap according to the instructions of the manufacturer.

The three participants were tested consecutively. The following factors were manipulated during testing. (1) Seat height of the subject: the normal individual seat height for knee exercises was defined as the position where the thigh is parallel to the floor and the lower leg in a 90° angle to the thigh. A 20% increase and decrease of subjects individual seating positions was used during testing. (2) Position of the BPMpathway sensor on the leg (transversal plane): according to the prescriptions of the manufacturer the correct position of the sensor for knee exercises in the sitting position is directly above the malleolus lateralis, perpendicular to the frontal plane of the shank. A 45° and a 90° ventral shift of the sensor were provoked additionally for testing. (3) Distance of the BPMpathway sensor to the joint: as mentioned above, the correct position of the sensor for knee exercises is directly above the malleolus lateralis. For test purposes the sensor was additionally placed in the mid of the shank and directly under the knee.

Each subject performed five movement repetitions in each task. The subjects were instructed to execute the movements at a self-selected constant speed.

### 2.5. Data Processing and Analysis

The data acquisition was carried out in both systems at a sample rate of 100Hz. The data of the BIOPAC MP160 was recorded with the software program AcqKnowledge^®^ 4.4 (BIOPAC Systems, Inc., California USA).

The manufacturer of the BPMpathway provided a special code, which allowed the logging of all the data recorded by the sensor. The angular values that were displayed on the screen during the test mode were stored as CSV files along with x/y/z Euler angles and max acceleration traces for synchronization with the data from the digital goniometer.

Both files were imported to MATLAB (MathWorks Inc., Natick, MA, USA). The synchronization of both files was made manually by detecting the change in baseline value where the subjects started to execute the test movements. A starting index in the datasets of both captures was identified and fixed. Each file contained the data of five full movement cycles (extension-flexion) (Figure 2).

Data analysis and visualization were carried out semi-automatically with the use of a MATLAB script developed for the current purpose.

Before the subjects started to perform the movement, the knee angle of the indexed leg was measured with an analog goniometer in the starting position. This angle was used in the script to define the offset of the BIOPAC dataset. Therefore, the first value of the BIOPAC dataset is exactly the value measured via the analog method.

Although the subjects were instructed to perform the movements at the same speed, the durations of the single movement cycles differ. For this reason, movement cycles were normalized in time (all cycles were interpolated with 100 samples over a period of one second). Afterwards, the five movement cycles were averaged in amplitude. The mean ± SD values for each system were plotted and the absolute difference of the mean curves was plotted as measurement error. A ROM mean difference of 5° between both systems was considered as clinically irrelevant. Since BIOPAC certifies an accuracy of ± 2° for the digital goniometer, a difference greater than 7° between both systems was considered as significant. When the ROM mean difference between both systems is zero, the error line touches the zero on the x-axis. Significant differences between the systems were defined as the time periods in which the error line goes above the 7° threshold (marked as a dashed line).

In addition to the graphical comparison of the data, the Root Mean Squared Error (RMSE) was used to calculate the difference between the values measured by the two devices. RMSE is always non-negative, and a value of 0 (almost never achieved in practice) would indicate a perfect fit to the data.

## 3. Results

The demographic data of the three subjects in the sample is presented in Table 1. Subjects B and C had normal BMIs but different statures. Subject A had a high BMI (overweight). To illustrate the results section, the data plots of subject C were used. The data plots of all three subjects in all tasks are presented in Appendix A.

### 3.1. Sitting Position

The seat height influenced the ROM data of the BPMpathway. When the subject sat in the prescribed position (normal seating position) and the sensor was placed according to the instructions of the manufacturer (above the malleolus lateralis) both systems showed similar knee angles during extension and flexion. However, the error line (difference between the means) sometimes crossed the 7° threshold (Figure 3a).

The RMSE for all three subjects in this position ranged from 5.9° to 15.5°, with subjects B and C showing similar results. A reduction of the prescribed sitting position by 20% led to significant ROM errors over the whole range of movement. While BIOPAC showed the correct knee angles, the BPMpathway displayed lower flexion angles and higher extension angles (Figure 3b). The RMSE increased to values up to 18.8°. A sitting height increase by 20% above the normal position also led to significant errors over the whole range of movement. Here, the BPMpathway displayed higher knee flexion and lower knee extension angles in comparison to the BIOPAC data (Figure 3c). The RMSE measured in this situation ranged from 8.8° to 13.4° (Table 2).

### 3.2. Position of the BPMpathway Sensor on the Transversal Plain (Ventral Shift)

The manipulation of the BPMpathway sensor position on the transversal plane influenced the ROM data displayed by the BPMpathway (Figure 4a,b). A 45° ventral shift of the BPMpathway sensor caused significant measurement errors during the extension movement (Figure 4a). The RMSE in this task ranged from 8.3° to 13.5°.

When the sensor was positioned in the frontal plane (90° shift to the prescribed position) significant errors were observed during the movement of extension, in extension and during the movement of flexion (Figure 4b). The RMSE ranged from 9.4° to 17°.

### 3.3. Position of the BPMpathway Sensor on the Sagittal Plain (Vertical Shift)

A vertical shift of the BPMpathway sensor was associated with a decrease of the measurement errors (Figure 5a,b). The absolute error between the mean curves of both instruments was marginal when the sensor was placed directly under the knee (Figure 5b). The RMSE was 3.2° and 2.4° for subject B and C respectively, with subject A showing a greater RMSE value (9°).

### 3.4. Body Composition of the Subjects

Despite inter-subject differences, the above-described effects of the manipulations (sitting height and sensor position) were consistent regardless of the body composition of the subjects, as shown in Table 2. While the RMSE values of subjects B and C were similar in the different tasks, the RMSE values of subject A (higher BMI) were lager, demonstrating greater differences between both systems.

## 4. Discussion

Knee ROM is an important clinical outcome that is related to patients’ physical function [24].

There are patient-related and intra-operatively modifiable factors that significantly predict the maximal knee flexion after TKA [25].

In the present study the effects of sitting position and sensor position on the accuracy of ROM data of an e-rehabilitation system were tested for knee exercises.

When the subjects sat in their normal position with sensor placement according to the instructions, the RMSE between the systems remained above the 5° threshold for all three subjects. The manipulation of the sitting position led to a decrease in accuracy with a correspondent increase in RMSE for all subjects. A transversal shift in the BPMpathway sensor position caused measurement errors too. According to the prescriptions of the manufacturer the sensor should be placed above the malleolus lateralis of the indexed leg for knee exercises. Our data showed that, against the prescription of the manufacturer, the accuracy of the knee angles measured by the e-rehabilitation system increased with a shift of the sensor to a position directly below the knee (Figure 4a–c). The lowest RMSE value of 2.4° was achieved in this position. These are the main findings of the present study.

ROM assessments after surgical knee interventions are mostly carried out with the use of an analogue goniometer by the clinician or physiotherapist at the clinic, during routine inspections or during outpatient physiotherapy. With the use of new technologies, it is possible to monitor knee ROM outside the clinical setting. In the past, different alternatives have been recommended to assess ROM with the use of smartphones [26,27,28] and sensors like gyroscopes, accelerometers, and magnetometers [29]. Some of the systems use two or three sensor units located above and below the indexed joint and sometimes in the trunk [30,31,32,33]. In the present study, we tested an e-rehabilitation system that determines the knee ROM based on the signals of a single sensor. The question whether the ROM data displayed by the BPMpathway system are comparable to the data obtained with a digital goniometer was investigated.

The data collected in the present study showed that the ROM data displayed by the BPMpathway system are sensible to factors like the sitting position of the subject while performing the exercise and the sensor position on the leg. The ROM data displayed by the system are calculated based on an algorithm. The system calculates the angles for knee exercises in sitting position based on the assumption that the subject is sitting with the thigh parallel to the floor. An increase or decrease in sitting height with the correspondent change of the relative position of the thigh in relation to the horizontal plane caused significant measurement errors.

Furthermore, the recommended sensor position for knee exercises should be reconsidered by the manufacturer. The lowest RMSE between the BPMpathway system and the concurrent system used (BIOPAC) was achieved, when the sensor was placed directly below the knee. This finding is probably related to the fact that one of the instruments inside the BPMpathway sensor is a three-axis accelerometer. Small movements of the shank during the flexion and extension movements are greater distally from the knee than directly below the knee, thus causing greater interferences.

The present preliminary study has limitations. One of the limitations is the small number of subjects in the sample. Another limitation relates to the fact that all tested subjects were healthy. The findings of the present investigation should be used to plan and carry out a concurrent validity study with a bigger sample size, in which the sensor should be placed directly under the knee for all knee exercises.

A feasibility study in a clinical setting with a small sample comprising patients of different BMIs should be carried out also, to assess the extent to which the system can be used in very early stages after TKA.

Although the e-rehabilitation system can also be used for hip and shoulder exercises, the results of the present study are only valid for knee exercises. The accuracy of the system for hip and shoulder exercises should be tested in a further study, since the recommendations for sensor positioning and the position of the subject during the exercises are joint specific.

## 5. Conclusions

The knee ROM data displayed by the BPMpathway system are comparable to the data of the concurrent system used in this investigation, provided the instructions of the manufacturer are respected concerning the sitting position of the subject for knee exercises, and disregarding the same instructions for sensor positioning on the leg by placing the sensor directly under the knee. RMSE values under 5° were achieved for two of the three subjects in the sample. Changes in seating position and rotation shifts of the sensor induced measurement errors.

## Figures and Tables

**Figure 1 sensors-20-02237-f001:**
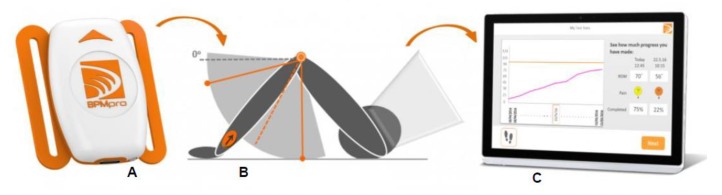
BPMPpathway sensor (**A**) that should be positioned above the malleolus lateralis (**B**) when the patient performs knee exercises. During the exercise the patients gets a live feedback on the actual performance over the patient application (**C**).

**Figure 2 sensors-20-02237-f002:**
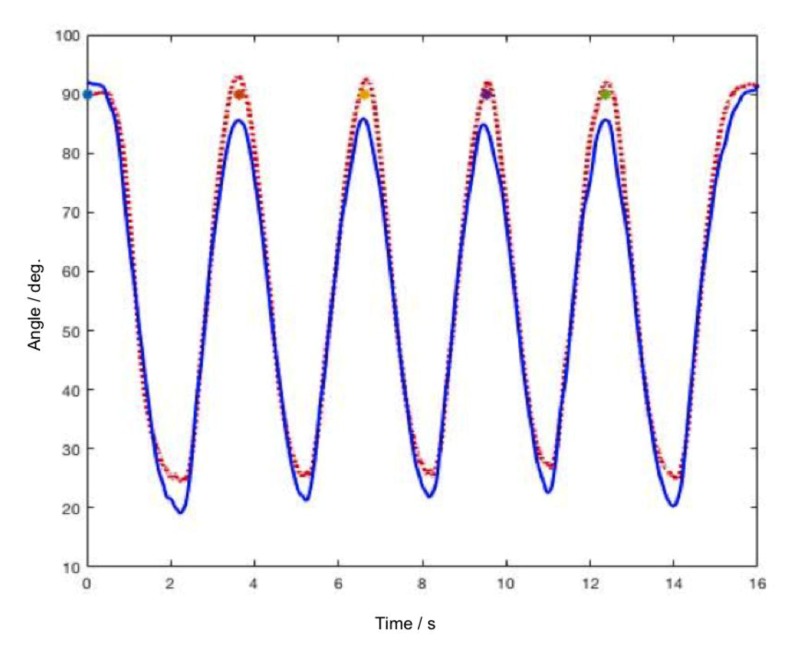
Raw data of five movement cycles in one of the tasks: BPMpathway (full blue line) and the BIOPAC goniometer (dotted red line).

**Figure 3 sensors-20-02237-f003:**
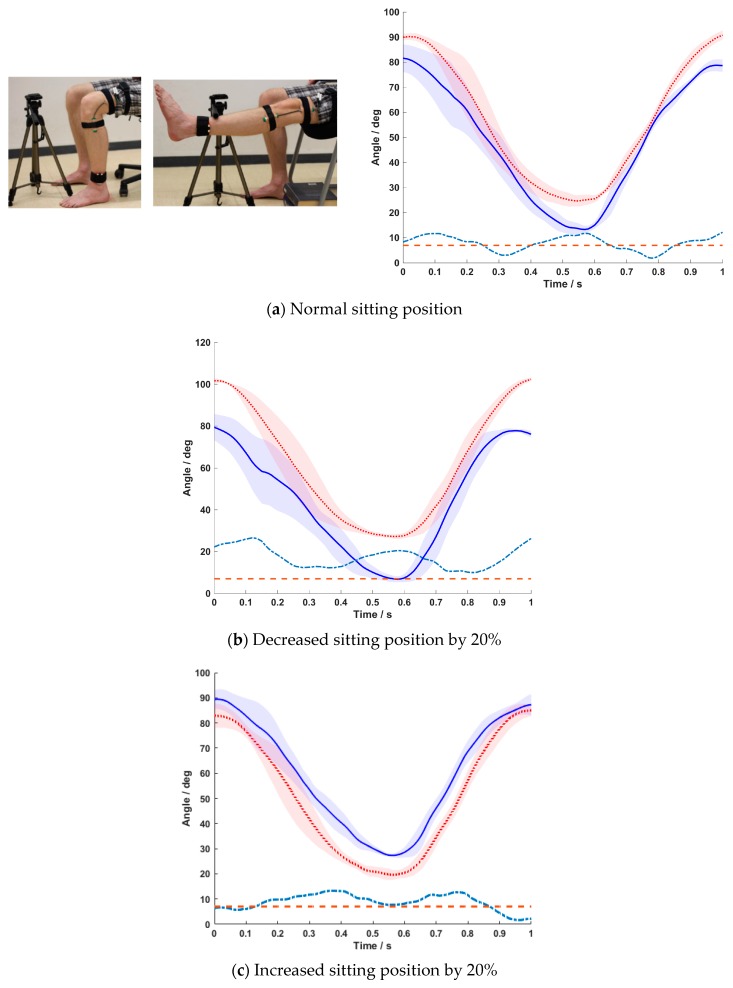
Mean ± SD of the five normalized movement cycles for the BPMpathway (full blue line) and the BIOPAC goniometer (dotted red line) with the difference between the means (dashed turquoise line) and the 7° tolerance threshold line (dashed red line) for normal sitting position (**a**), 20% lower sitting position (**b**), and 20% higher sitting position (**c**).

**Figure 4 sensors-20-02237-f004:**
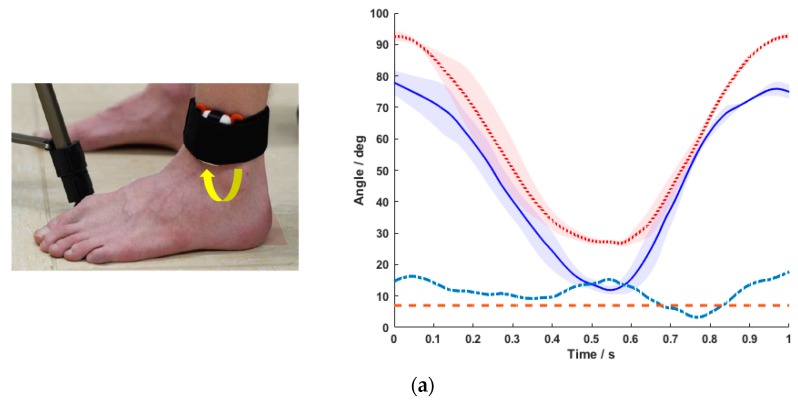
Mean ± SD of the five normalized movement cycles for the BPMpathway (full blue line) and the BIOPAC goniometer (dotted red line) with the difference between the means (dashed turquoise line) and the 7° tolerance threshold line (dashed red line) for sensor positioning 45° ventrally (**a**), and 90° ventrally from the prescribed position (**b**).

**Figure 5 sensors-20-02237-f005:**
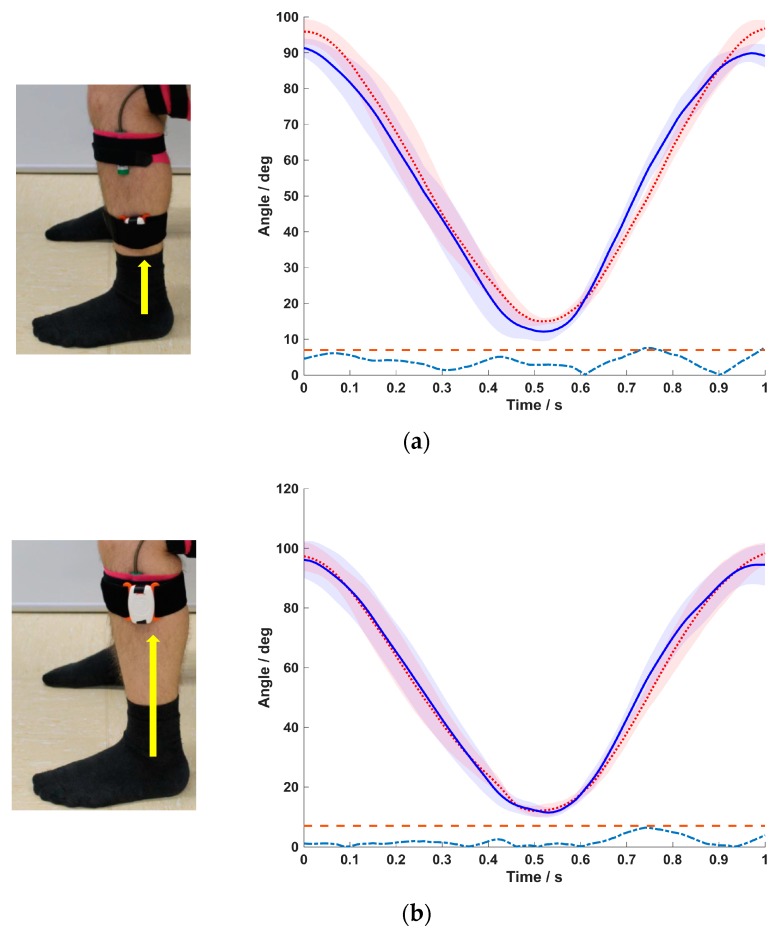
Mean ± SD of the five normalized movement cycles for the BPMpathway (full blue line) and the BIOPAC goniometer (dotted red line) with the difference between the means (dashed turquoise line) and the 7° tolerance threshold line (dashed red line) for normal sitting position with sensor at the mid of the shank (**a**) and sensor directly under the knee (**b**).

**Table 1 sensors-20-02237-t001:** Demographic data of the participants.

Subject	Age	Body Height (cm)	Body Weight (kg)	BMI (kg/m²)
A	28	185	124	36.2
B	24	172	72.4	24.5
C	26	187	76.2	21.8

**Table 2 sensors-20-02237-t002:** Root Mean Squared Errors (RMSE) for all subjects in each test situation.

		Subject A	Subject B	Subject C
Sitting position	Normal	15.5	5.9	8.3
	20% lower	18.8	17.8	17.9
	20% higher	13.4	8.8	9.3
Sensor position	Rotate 45°	8.3	7.1	11.8
	Rotate 90°	17.0	9.4	13.1
	Mid shank	9.5	6.4	4.2
	Below knee	9.0	3.2	2.4

RMSE values are in degrees.

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
