# Peer review of "Sensor Positioning Influences the Accuracy of Knee Rom Data of an E-Rehabilitation System: A Preliminary Study with Healthy Subjects"

_sensors, 2020, doi:10.3390/s20082237_

Round 1

Reviewer 1 Report

The authors should be commended for this concise and clearly written paper. Although the paper may appear quite simple it is impactful and I can see many clinicians benefiting from this output. Indeed, there are some limitations particularly with the small number of participants. However, this is a preliminary study with a lot of scope for future studies incorporating various patient populations with musculoskeletal and neuromuscular deficits. As a potential rehabilitation tool clinicians would find the outcome of this study very useful and beneficial to their healthcare centres and clinical practices.

There are a few minor spelling mistakes and grammatical errors.

In most sections please refrain from making one sentence paragraphs.

A stronger background supported with a rationale and justification for carrying out the study is needed here. This should build up to possibly highlighting a research question or gap identification in the research. Why is this study important? What's novel about this this study?

Why don't you have a study purpose and hypothesis at the end of your introduction.

Line 60: Spelling 'Preliminary'.

Line 81: Clarity required in the sentence. Change to 'This ensures the progress and recovery of the patient is monitored'.

Good figures. Labe axes in Figure 2.

Line 131: Insert a comma between text ( ...movement, the knee...)

Line 133: Be careful with your abbreviation 'BIOPAC' not 'BIOAC'

Line 203: Insert 'of' in sentence  (...regardless of the body composition..)

Line 204: Delete 'visible ' and replace with 'shown'

Author Response

Reviewer #1

The authors would like to thank reviewer #1 for his/her critical feedback, which helped us to improve the manuscript.

There are a few minor spelling mistakes and grammatical errors.

  • The authors thank reviewer #1 for the correction tips, which were all implemented.

In most sections please refrain from making one sentence paragraphs.

  • Ahead of submission the manuscript was proof read by a native English speaker. We think the paragraphs are written in an understandable way.

A stronger background supported with a rationale and justification for carrying out the study is needed here. This should build up to possibly highlighting a research question or gap identification in the research. Why is this study important? What's novel about this this study?

Why don't you have a study purpose and hypothesis at the end of your introduction.

  • We agree with reviewer #1, the rationale for the study was not highlighted in the introduction section of the manuscript. We added a last paragraph to the introduction with the aim of the study. This last paragraph highlights the rationale for carrying out the study.

Line 60: Spelling 'Preliminary'.

  • We apology for the typing error that we have corrected now.

Line 81: Clarity required in the sentence. Change to 'This ensures the progress and recovery of the patient is monitored'.

  • We have changed the text according to the suggestion of reviewer #1.

Good figures. Labe axes in Figure 2.

  • We have labeled the axes of Figure 2 according to the suggestion of reviewer #1.

Line 131: Insert a comma between text ( ...movement, the knee...)

  • A comma was added to the sentence according to the suggestion.

Line 133: Be careful with your abbreviation 'BIOPAC' not 'BIOAC'

  • Thank you for the hint. The typing error was corrected.

Line 203: Insert 'of' in sentence (...regardless of the body composition..)

  • The correction was done according to the suggestion.

Line 204: Delete 'visible ' and replace with 'shown'

  • The correction was done according to the suggestion of reviewer #1.

Reviewer 2 Report

Well planned and done. Best wishes.

Please see detailed comments in the attachment.

Author Response

Reviewer #2

Well planned and done. Best wishes.

The authors would like to thank reviewer #2 for his/her critical feedback.

In the attached pdf-file of the manuscript reviewer #2 made two comments. Both comments concerned the introduction section of the manuscript:

1st comment: We have deleted the words “short term” from the sentence and have added 3 further references to reinforce the importance of physiotherapy and exercise after THA and TKA. The sentence runs now as follows:

“After TKA and THA physiotherapy and exercise lead to improvements in physical function [5-8].”

The three new references added to the manuscript are the following ones:

  1. Bandholm, T.; Kehlet, H., Physiotherapy exercise after fast-track total hip and knee arthroplasty: time for reconsideration? Arch Phys Med Rehabil 2012, 93, (7), 1292-4.
  2. Di Monaco, M.; Vallero, F.; Tappero, R.; Cavanna, A., Rehabilitation after total hip arthroplasty: a systematic review of controlled trials on physical exercise programs. Eur J Phys Rehabil Med 2009, 45, (3), 303-17.
  3. Henderson, K. G.; Wallis, J. A.; Snowdon, D. A., Active physiotherapy interventions following total knee arthroplasty in the hospital and inpatient rehabilitation settings: a systematic review and meta-analysis. Physiotherapy 2018, 104, (1), 25-35.

2nd comment: “(…) seem not to be inferior to standard physiotherapy [12-17].”

Please describe shortly but precisely in which aspects?

  • We have added additional information to the sentence. The sentence runs now as follows: “(…) seem not to be inferior to standard physiotherapy in terms of improvements in range of movement and muscle strength, decrease in pain, patient satisfaction and patient reported outcome measures (PROMs) [15-20].”